# High Risk, High Dose?—Pharmacotherapeutic Prescription Patterns of Offender and Non-Offender Patients with Schizophrenia Spectrum Disorder

**DOI:** 10.3390/biomedicines10123243

**Published:** 2022-12-13

**Authors:** Lena Machetanz, Moritz Philipp Günther, Steffen Lau, Johannes Kirchebner

**Affiliations:** 1Department of Forensic Psychiatry, University Hospital of Psychiatry Zurich, 8032 Zurich, Switzerland; 2Department of Consultation-Liaison Psychiatry and Psychosomatic Medicine, University Hospital Zurich, University of Zurich, 8006 Zurich, Switzerland

**Keywords:** schizophrenia spectrum disorders, antipsychotics, polypharmacy, overdosing, offender patients, forensic psychiatry, benzodiazepines, antidepressant

## Abstract

Compared to acute or community settings, forensic psychiatric settings, in general, have been reported to make greater use of antipsychotic polypharmacy and/or high dose pharmacotherapy, including overdosing. However, there is a scarcity of research specifically on offender patients with schizophrenia spectrum disorders (SSD), although they make up a large proportion of forensic psychiatric patients. Our study, therefore, aimed at evaluating prescription patterns in offender patients compared to non-offender patients with SSD. After initial statistical analysis with null-hypothesis significance testing, we evaluated the interplay of the significant variables and ranked them in accordance with their predictive power through application of supervised machine learning algorithms. While offender patients received higher doses of antipsychotics, non-offender patients were more likely to receive polypharmacologic treatment as well as additional antidepressants and benzodiazepines. To the authors’ knowledge, this is the first study to evaluate a homogenous group of offender patients with SSD in comparison to non-offender controls regarding patterns of antipsychotic and other psychopharmacologic prescription patterns.

## 1. Introduction

The legal system and criminal law in most jurisdictions in the western world provide for forensic psychiatric care for offenders who are deemed to not be accountable for a committed offense due to a psychiatric disorder [1]. This means that treatment not only addresses the patient’s well-being and individual needs, but also serves to protect society from criminal recidivism caused or negatively influenced by said underlying psychiatric disorder [2,3]. In Switzerland, the legal basis for such inpatient forensic therapy is provided by Article 59 of the Swiss Penal Code, and treatment is carried out in forensic psychiatric institutions, residential facilities, or in specialized prison departments [4]. Despite the generally accepted view that criminal behavior stemming from psychiatric disorders is preventable through treatment, there is a shortage of knowledge on pharmacological treatment in forensic psychiatry [4,5]. Findings from general psychiatry cannot be directly applied to forensic psychiatry due to a variety of systematic differences between patient populations and treatment circumstances, including the higher proportion of comorbidity, the compulsory context of treatment due to court mandated therapy, and the history of severe violence [6,7,8,9]. Compared to acute or community settings, forensic psychiatric settings in general have been reported to make greater use of antipsychotic polypharmacy and/or high dose pharmacotherapy, including overdosing [10,11,12,13,14]. However, previous findings on offender patients have mainly been based on mixed populations with different psychiatric diagnoses, and, although they constitute most inpatient forensic psychiatric patients, offender patients with schizophrenia spectrum disorders (SSD) have not been thoroughly evaluated as their own entity [15,16,17,18,19,20]. One recent study by Günther et al. has explored characteristics of offender patients who are most likely to receive high dose antipsychotic (poly-)pharmacy, but this study lacked a non-offender control [21].

Therefore, our first objective was to test the following hypotheses:

**Hypotheses I.** 
*Offender patients receive higher doses of antipsychotic drugs.*


**Hypotheses II.** 
*Offender patients are more often subjected to antipsychotic polypharmacy.*


**Hypotheses III.** 
*Offender patients receive benzodiazepines for sedation more frequently.*


Our second objective was to evaluate the interplay between the variables found to be of significance, and to test the performance measures of a model distinguishing between forensic and non-forensic offender patients with SSD. 

## 2. Materials and Methods

This study was reviewed and approved by the Ethics Committee of Zurich. Initially, as our study group, we chose 370 male and female offender patients with a diagnosis of F2x according to ICD-10, who had all been in court-mandated treatment at the Centre for Inpatient Forensic Therapies of the University Hospital of Psychiatry, Zurich, Switzerland [22]. Offenses leading to the referenced forensic psychiatric hospitalization included both violent crimes—(attempted) homicide, assault, violent offenses against sexual integrity, robbery, and arson—and non-violent crimes—threat and coercion, property crime without violence, criminal damage, traffic offenses, drug offenses, and illegal gun possession. The offender population has been used for explorative analyses as part of a larger project aiming at evaluating the complexities of offender patients with SSD, and further information on data collection regarding this overall population can be found in in Lau et al. [23]. The control group was composed of 370 non-offender patients suffering from SSD (F2x also acc. to ICD-10), who had been in treatment at the Centre for Integrative Psychiatry of the University Hospital of Psychiatry, Zurich [22]. The facility is an inpatient institution with a rehabilitative focus and a patient population mostly comprising of patients affected by chronic and/or prolonged courses of disorder. This control group was chosen because—as for forensic psychiatric patients—an initial treatment for acute psychosis had already been established in most cases (either on an acute psychiatric ward or in a prison setting), as well as due to its large portion of chronically ill patients.

Both the offender and non-offender patients were matched according to age and gender. As we aimed to evaluate parameters linked to pharmacologic treatment, as well as to consider the pharmacological innovations of the last decades, especially partial dopamine agonists which were introduced in Switzerland in 2004, we included only patients from 2005 to 2016. Patients with an admission date earlier than 2005 were excluded in both groups. Our final sample then consisted of 178 non-offender patients and 206 offender patients. Amongst the latter, 40.4% had committed a violent offense, including violent crimes against sexual integrity, which accounted for 2.4% of all violent offenses. The mean age at the first entry into the Swiss criminal registry was 24.7 years. The majority of the total population had a diagnosis of paranoid schizophrenia (F20.0 acc. to ICD-10), with a rate of 79.5% in the non-offender group and 86.9% in the offender group. The remaining percentage comprised schizoaffective disorders, hebephrenic schizophrenia, and acute psychotic disorder.

Data from the files of these patients were retrospectively assessed in a structured manner, applying a directed qualitative content analysis [24]. This content analysis was performed according to a rating protocol for coding based on a set of criteria originally proposed by Seifert et al., and adapted in inter- and supervisions with senior researchers in (forensic) psychiatry [25]. The case files were comprehensive, and included professionally documented anamneses, psychiatric/psychologic inpatient and outpatient reports, extensive reports from clinicians as well as nursing and care staff, and, where applicable, testimonies, court proceedings, and data regarding previous imprisonments and detentions. Therefore, the dataset consisted of items from the following domains: social-demographic data, childhood/youth events, psychiatric history, past criminal history, social/sexual functioning, prison data, and particularities of the current hospitalization and psycho-pathological symptoms defined by an adapted three-tier positive and negative syndrome scale (PANSS). To allow for comparability of dosages of the different antipsychotics, dose data were converted to olanzapine equivalents. This was either achieved through the classical weighted mean dose method and the minimum effective dose method, or, wherever these were not possible, based on international expert consensus [26,27,28]. For a detailed description and definition of all predictor variables, please refer to our coding protocol in the data availability statement.

We performed an independent sample Mann–Whitney U-Test for all metric variables with non-normal distribution, and a Fisher’s exact test for all other variables [29,30]. Adjustment for alpha error was performed using the Benjamini and Hochberg Method [31]. The level of significance was defined as *p* < 0.05.

In the second step, we aimed to evaluate the interplay of the variables and to rank them in accordance with their influence on the model. Furthermore, we wanted to evaluate the performance of said model regarding its ability to differentiate between the two groups, going beyond the *p*-value. For this purpose, we applied supervised machine learning (ML). Parts of the following section were already published in another study from our research group and are partly replicated here due to our use of the same methodology [32]. overview of the statistical steps is shown in Figure 1 and is further described in detail below. 

All of the steps were performed using R version 3.6.3. (R Project, Vienna, Austria) and the MLR package v2.171 (Bischl, Munich, Germany). Calculations of the balanced accuracy were conducted using MATLAB R2019a (MATLAB and Statistics Toolbox Release 2012, The MathWorks, Inc., Natick, MA, USA) with the add-on “computing the posterior balanced accuracy” v1.0. All raw data were first processed for ML (see Figure 1, Step 1), and several categorical variables were converted to binary code. Continuous and ordinal variables were not adjusted. The independent variable was dichotomized into (a) “offender patients” and (b) “non-offender patients“. An elimination of variables due to missing values of 30% or more was not indicated, as all variables had <10% of missing values, except for “*regular intake of antipsychotic medication*” (missing values: 29.7%). After data preparation, the database was divided into one training subset containing 70% of all cases, and one validation subset containing the remaining 30% (see Figure 1, Step 2). The training subset was used for variable reduction and model building/selection. To enable the flexible application of all ML algorithms, imputation of missing values was carried out by mean for continuous variables and by mode for categorical variables. Imputation weights were saved for later to be reused on the validation subset (see Figure 1, Step 3a). As we aimed to identify the most influential variables, and as a decrease in variables could counteract overfitting while maintaining computing times in initial model building at an acceptable level, we performed a variable reduction through randomForestSRC, down to the point where the AUC improved by no more than 5% through adding another item (see Figure 1, Step 3b). This led to a variable reduction to the 8 most predictive variables. With the database of *n* = 384 being relatively small for ML purposes, we applied discriminative model building with logistic regression, trees, random forest, gradient boosting, KNN (k-nearest neighbor), and support vector machines (SVM), as well as, for easily applicable generative model building, naïve Bayes (see Figure 1, Step 3c). For each model, performance was calculated and assessed in terms of its balanced accuracy (the average of true positive and true negative rate, better suited for model evaluation and calculation of confidence intervals in imbalanced data) and goodness of fit (measured with the receiver operating characteristic, balanced curve area under the curve method, and ROC-balanced AUC). Specificity, sensitivity, positive predictive value (PPV), and negative predictive value (NPV) were also evaluated. The model with the highest AUC was then chosen for final model validation with the test subset (see Figure 1, Step 3d). As our sample size was relatively small, we were careful about avoiding overfitting, a common obstacle in ML occurring when, e.g., outliers are incorporated into the model. For this purpose, it is advisable to apply imputation, variable reduction, and model building in a cross-validation process, and to keep this separate from the testing of the model. In our study, a nested resampling approach was employed, using a nested resampling model with the inner loop performing imputation, variable filtration, and model building within five-fold cross-validation. The outer loop, for performance evaluation, was also embedded in five-fold cross-validation (see Figure 1, Step 4). Through cross-validation, five different equally-sized subsamples of our dataset were artificially created, allowing one subset to serve as training set for our model, while the remaining four subsets allowed for the evaluation of the accuracy of the learned model [33,34]. To evaluate the previously selected model, we applied the validation subset, which included 30% of all cases (see Figure 1, Steps 5–7). The previously stored imputation weights were reused for the validation subset (see Figure 1, Step 5). Then, the selected model was applied for validation (see Figure 1, Step 6). The identified variables were finally ranked according to their indicative power (see Figure 1, Step 7).

## 3. Results

### 3.1. Null Hypothesis Significance Testing (NHST)

The absolute and relative distributions of all predictor variables in the NHST, as well as their levels of significance, are shown in Table 1. 

#### 3.1.1. Hypothesis I: Offender Patients Receive Higher Doses of Antipsychotic Drugs

With a *p*-value of 0.000, offender patients received significantly higher doses of antipsychotics. This applied to both admission (21.4 mg vs. 14.6 mg) and discharge (22.1 mg vs. 19.3 mg). Hypothesis I can, therefore, be confirmed.

#### 3.1.2. Hypothesis II: Offender Patients Are More Often Subjected to Antipsychotic Polypharmacy

Upon their admission, 23.4% of the offender patients were treated with two or more antipsychotic substances, while non-offenders were subjected to polypharmacy in 37% of all cases. With a *p*-value of 0.024, this result was significant. Upon their discharge, there was no significant difference regarding polypharmacy between the two groups. Hypothesis II, therefore, needs to be rejected.

#### 3.1.3. Hypothesis III: Offender Patients Receive Benzodiazepines for Sedation More Frequently

An additional prescription of benzodiazepines occurred in 18.1% of all offender patients upon their admission, and 38.2% of all non-offender patients. With a *p*-value of 0.000 each, these differences were significant. Consequently, hypothesis III needs to be rejected. In addition, offender patients did significantly receive less augmentation with antidepressants than their non-offender controls (8.8% v. 35.4%).

### 3.2. Model Calculation Using Machine Learning (ML)

Variables were introduced into seven different ML algorithms, with the “length of stay” variable being omitted. The reason for this was that the length of stay was determined by structural differences, with court-mandated therapy possibly lasting 5 years or longer according to its legal ground, while this is rarely the case for general psychiatric hospitalizations. With a balanced accuracy of 77.8% and an AUC of 0.87, support vector machines (SVM) outperformed all other algorithms, and were, therefore, identified as the most suitable (see Table 2). 

Table 3 provides an overview of the performance measures of the final SVM model. In this final model, the AUC yielded 73.7%, with a Balanced Accuracy of 0.83. With a sensitivity of 67%, offender patients could be correctly identified in 7 out of 10 cases based on the eight predictor variables. Non-offender patients were identified correctly in eight out of ten cases (specificity of 82%). 

### 3.3. Ranking of Predictor Variables

Figure 2 shows the ranking of all eight predictor variables according to their relative influence in the final validation model. 

The olanzapine equivalent at the time of discharge from the referenced hospitalization was identified as most predictive variable, closely followed by regular intake of antipsychotic medication and the additional prescription of an antidepressant. Further predictors included the olanzapine equivalent at the time of the admission to the referenced hospitalization, the additional prescription of benzodiazepines, previous antipsychotic medication, and previous out- and inpatient treatment. Polypharmacy, however, while highly significant in the NHST, did not prove to be among the top eight predictive variables in distinguishing offender from non-offender patients with SSD. 

## 4. Discussion

Schizophrenia spectrum disorders (SSD) are severe mental disorders with a heterogeneous combination of symptoms and a lifetime prevalence of around one percent [35,36]. Frequently, affected patients are unable to cope with the challenges of their day-to-day life, and experience impairment and disability in multiple domains, including the ability to maintain social relationships, sustain employment, and live independently [37,38]. Additionally, the diagnosis of SSD is associated with a substantially higher risk of violent behavior and committing a violent crime [5,9,39,40]. However, most of the patients suffering from SSD do not show aggression or criminal behavior. Well-established risk factors for violence in SSD include a comorbidity of substance use disorders, a history of hostile behavior, and non-adherence to pharmacotherapy [9,40]. Even though criminal behavior can be considered preventable through adequate treatment of the underlying SSD, there is a scarcity of research on pharmacologic treatment as the central pillar of SSD therapy in offender populations. As outlined in the introduction, psychiatric findings from non-offender patients with SSD cannot automatically be applied to offender populations with SSD due to numerous systematic differences, e.g., coercive therapy context, duration of treatment, and history of severe aggression [41]. The aim of our study was, therefore, to examine key differences in the treatment of offender and non-offender patients with SSD, investigating a sample of 206 offender patients from a forensic psychiatric institution and a matched control of 178 non-offender patients from a rehabilitative psychiatric institution, both of which were admitted for long-term treatment. We hypothesized that offender patients were more likely to receive higher doses of antipsychotics (H I) and antipsychotic polypharmacy (H II), as well as benzodiazepines more frequently than their non-offender comparisons (H III). Furthermore, we intended to evaluate the interplay of the variables which most significantly distinguished between the two groups through the application of suitable machine learning algorithms. 

With a mean olanzapine equivalent of 21 mg at admission and 22 mg at discharge, offender patients had a significantly higher dose of antipsychotic medication at both measurements—this was especially striking as the psychological severity, measured by the PANSS, did not significantly differ between the two groups. Consequently, hypothesis I can be confirmed. However, non-offender patients had a higher likelihood of having an additional benzodiazepine and antidepressant prescription, and were significantly more often subjected to polypharmacy upon their admission to the referenced hospital. Therefore, hypotheses II and III are to be refuted. The influence of factors unrelated to psychopathology on decisions regarding polypharmacy and antipsychotic dosing has been brought up before. Günther et al. found emotional rapport, withdrawal, and the absence of a comorbid personality disorder to increase the odds for high-dose antipsychotic treatment [21]. Stone-Brown et al. have also described higher doses of antipsychotics and less polypharmacy in psychiatric offender populations than in general psychiatric populations [42]. While it seems surprising that the offender population, which may be considered more prone to impulsivity and agitation, received fewer benzodiazepine prescriptions than sedative substances, a reason for conservative benzodiazepine prescription could lie in the greater awareness of possible paradoxical reactions and provocation of addiction in a population with an already higher burden of substance use disorders. Although relatively uncommon, with a prevalence of less than 1% in all psychiatric patients, there is evidence that paradoxical reactions with excitement and agitation instead of sedation are more likely to occur in patients with a history of aggressive and violent behavior and alcohol use disorders, making the offender population especially vulnerable [43]. Another possible explanation is that patients who received greater doses of antipsychotics were not in need of further sedative substances such as benzodiazepines, as they already experienced sedation through the side effects of their antipsychotic treatment. The rarer use of antidepressant agents in the offender population may be explained by clinicians’ suspicion of re-exacerbation of positive psychotic symptoms under antidepressant pharmacotherapy [44]. Another common worry amongst clinicians is the increased rate of side effects when combining antipsychotics and antidepressants due to higher plasma levels of both substances, resulting from competitive inhibition of hepatic microsomal oxidative enzymes [45]. Yet, adjunctive antidepressants have been shown to be beneficial to patients with SSD regarding negative symptoms, with manageable risk of psychotic exacerbation [46,47] is, however, little knowledge on which subgroups of offender patients with SSD benefit most from such an augmentation.

Further significant differences between offender and non-offender patients emerged, which shall be discussed in the following sections.

Due to the matching of the two samples, the groups did not differ in age at admission nor in gender, both being predominantly male, in their mid-thirties, and, as described above, similar in their diagnostic composition, which allowed for good comparability. However, offender patients had significantly lower levels of education, and were significantly more often born outside of Switzerland. The correlation between lower levels of education and violence among patients suffering from SSD has been described in previous trials as well [48,49,50]. While research on a possible correlation between SSD in migrants and violent behavior shows inconsistent results, it can be argued that cultural and language barriers may prevent patients with SSD from accessing the mental health care and support systems provided in Switzerland, and may also complicate diagnostic and therapeutic processes [40,51,52]. This hypothesis matches our next finding: regarding their psychiatric history, offender patients were shown to be less likely to have had in- or outpatient treatments, or to have had an antipsychotic prescription before the referenced hospitalization. In comparison to their non-offender controls, they were also less likely to regularly take a prescribed antipsychotic medication. These findings are in line with previous findings that lower levels of treatment adherence are known to correlate with higher rates of violent and non-violent offending [53,54]. Patients who are well-embedded in the mental health system may be more aware of their diagnosis and need for treatment. Close contact with healthcare professionals also enables early intervention regarding prevention of the deterioration of mental health and, possibly, consecutive violent behavior, e.g., through adaptation of pharmacotherapy or alerting of appropriate protective agencies. 

In turn, the offender population had significantly higher rates of comorbidities of both alcohol and substance use disorders. This comes as no surprise, as substance use comorbidities are well known to gravely amplify the risk of violent behavior in SSD patients [9,40,55]. While the two samples did not differ regarding comorbid personality disorders, case numbers in both groups were too low to infer robust results (n = 31 vs. n = 14). The small prevalence in our population can be explained, as clinicians are reluctant to diagnose a comorbid personality disorder in patients suffering from SSD. This is in accordance with the ICD-10, as personality disorders may only be diagnosed once other severe mental disorders which could explain the symptomatology are ruled out [22].

Offender patients also had a significantly higher rate of compulsory measures during their referenced hospitalization than the non-offender controls, including isolation, compulsory medication, and restraint. On the one hand, this seems logical, as a history of aggression, which offender patients usually have, is known to be a risk factor for further aggressive behavior both in and outside of psychiatric institutions [40]. However, this may not necessarily mean that they also show a higher rate of inpatient aggression compared to non-offenders. A recent study by our research group investigating inpatient aggression showed that only a third of all offender patients showed violent behavior during their referenced forensic institutionalization [56]. Compared to this, the prevalence of aggressive behavior in general psychiatric wards ranges between 15% and 53% for patients with SSD [57]. However, compulsory measures can not only be necessary in cases of endangerment of others, but also in cases of self-harm, which occurs at a higher rate in forensic inpatient settings than it does in general psychiatric settings (42.9% vs. 17.4%) [58]. Another factor possibly contributing to the increased rates of compulsory measures in the offender population may be a lower threshold for coercive interventions in the treatment of patients who are known to be particularly prone to violent and impulsive behavior. 

The significantly longer length of stay in the offender population is easily explicable through the circumstances of admission: patients admitted to a forensic psychiatric facility according to Article 59 of the Swiss Penal Code are to receive court-mandated treatment, which can last for up to 5 years as a first step, and this may even be prolonged afterwards if the offender’s risk of future offenses has not yet been sufficiently reduced [59]. The requirements to release offender patients from inpatient treatment are obviously much stricter than for SSD patients who have not proven to be dangerous to society. As briefly described in the Methods section, we, therefore, chose to omit this item for further analysis with ML. Otherwise, this variable would have dominated the entire model, even though it would only have been an expression of a structurally determined difference.

When ranking the identified predictors in accordance with their contributed weight to the ML model, the olanzapine equivalent at the time of discharge from the referenced hospitalization, as a measure of the cumulative antipsychotic dose, emerged as the most relevant factor in distinguishing offender and non-offender patients. The following five most powerful predictors also referred to psychopharmacologic treatment (regular intake of antipsychotic medication, additional prescription of antidepressants and benzodiazepines, and a history of antipsychotic pharmacotherapy). While the two groups also significantly differed regarding the prevalence of psychiatric comorbidities, neither of these variables emerged as nearly as powerful regarding their predictive value in the model as pharmacological items—which is underlined by the fact that the AUC of the model did not improve by more than 5% through adding another item. The observation that prescription rather than clinical features by far dominated the model could be an expression of the different treatment settings, approaches, and goals between forensic and general psychiatric institutions.

Summing up the key differences between offender and non-offender patients with SSD, the following emerged: for the offender patient, the results paint a picture of a patient burdened with more substance-related comorbidities, less integration into the therapeutic system, and less medication compliance, who is subsequently subject to more frequent restraints and consistently higher doses of antipsychotic medication during his course of treatment. Conversely, the non-offender patient presents as a better therapeutically covered and more compliant patient with less of a burden of additional psychiatric diagnoses, who more often receives sedative and antidepressant drugs in addition to a generally lower level of antipsychotic medication. In the ML model, parameters related to pharmacotherapy emerged as most able to distinguish between offender and non-offender patients with SSD, and dominated factors regarding comorbidities and symptomatology. This implies that the two groups differ more in their treatment than in their clinical presentation, which seems understandable given the differing treatment settings and desired outcomes between general and forensic psychiatry. 

Considering limitations, the most serious is that the present study was based exclusively on retrospective data collection. While we aimed at ensuring a sufficient quality of data by using a structured data extraction protocol, this does not meet the data quality of a prospectively standardized study. Furthermore, because the treatment decisions were not evaluated, as could have been conducted in a prospective design, we can only speculate about the treating physicians’ rationale for their choices of dosage and substance. Further research is needed to examine treatment decisions in forensic psychiatric populations and to find out whether they differ from those made in general psychiatry. On the other hand, for example, with dosage, previous in- and outpatient behavior, or prescription of additional psychopharmaceutical substances, most of our predictor variables were robust items with little to no possibility of different interpretations between raters. Another limitation that is often found in forensic psychiatric research was the small population, especially when compared to populations in medical research in other specialties using machine learning, where samples are often composed of several thousands of cases. This aggravates overfitting, which is a common obstacle in ML, and which we have tried to limit through a nested resampling approach which we applied to every step in the ML process. The relatively small sample made inferences in certain aspects difficult, for example, regarding the comorbidity of personality disorders, as the case numbers with said comorbidity were too small in both subgroups. In addition, while the majority of the patients were diagnosed with paranoid schizophrenia (ICD-10: F20.0), one needs to address a certain diversity amongst the population regarding their diagnoses. Nevertheless, due to the heterogeneity of disorders from the schizophreniform spectrum, we decided not to exclude the other diagnoses from the population, especially in view of the abandonment of subtypes of schizophrenia in ICD-11 [60]. Lastly, as the population was predominantly male, with <10% females, the applicability to women in forensic psychiatric institutions and generalizability of the results are limited. To draw robust causal conclusions, there is, therefore, a need for a reproduction of this study in a larger patient population, with the hope to include more women as well. 

## 5. Conclusions

To the authors’ knowledge, this is the first study to evaluate a homogenous sample of offender patients with SSD regarding their prescription patterns in comparison to non-offender controls. While offender patients seem to be burdened with more comorbidities and to be less integrated in a therapeutical setting, as well as to be less adherent to pharmacological treatment, they also tend to receive a smaller variety of psychopharmacologic agents, including benzodiazepines and antidepressants. This highlights the importance of early and sufficient integration of patients suffering from SSD into the mental health system, especially those who show signs of violent behavior. The fact that all variables with the most predictive influence in distinguishing offender from non-offender patients were related to pharmacotherapeutic aspects, instead of factors associated with symptomatology, could be an expression of the different treatment settings, approaches, and goals between forensic and general psychiatric institutions. While the authors advocate for a prescription policy based on objective parameters in forensic psychiatry, such as symptomatology scores (e.g., PANSS) to avoid unnecessary antipsychotic overdosing, this study did not evaluate decision-making regarding prescriptions. Further research should, therefore, focus on examining the underlying rationale for individual treatment decisions, and exploring whether there are subgroups of offender patients that could benefit from augmentative pharmacotherapeutic strategies. 

## Figures and Tables

**Figure 1 biomedicines-10-03243-f001:**
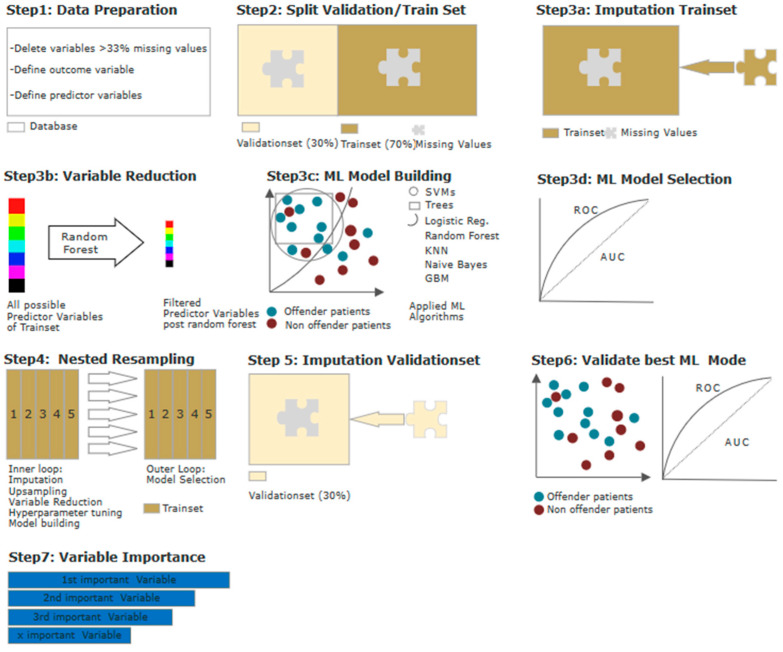
Machine learning model building and validation, step by step: Step 1—Data preparation: Multiple categorical variables were converted to binary code. Continuous and ordinal variables were not manipulated. Outcome variable offender/non-offender patient and predictor variables were defined. Step 2—Datasplitting: Split data into 70% training dataset and 30% validation dataset. Step 3a–d—Model building and selection: (a) imputation by mean; (b) variable reduction via random forest; (c) model building via ML algorithms—logistic regression, trees, random forest, gradient boosting, KNN (k-nearest neighbor), support vector machines (SVM), and naïve Bayes; and (d) testing (selection) of best ML algorithm via ROC parameters. Step 4—Model building and testing on training data: nested resampling with imputation, variable reduction, and model building in inner loop, and model testing on outer loop. Step 5—Imputation with stored weights from Step 3a on validation set. Step 6—Model building and testing on validation data: best model identified in Step 3c applied on imputed validation dataset and evaluated via ROC parameters. Step 7—Test for multicollinearity and ranking of variables by indicative power.

**Figure 2 biomedicines-10-03243-f002:**
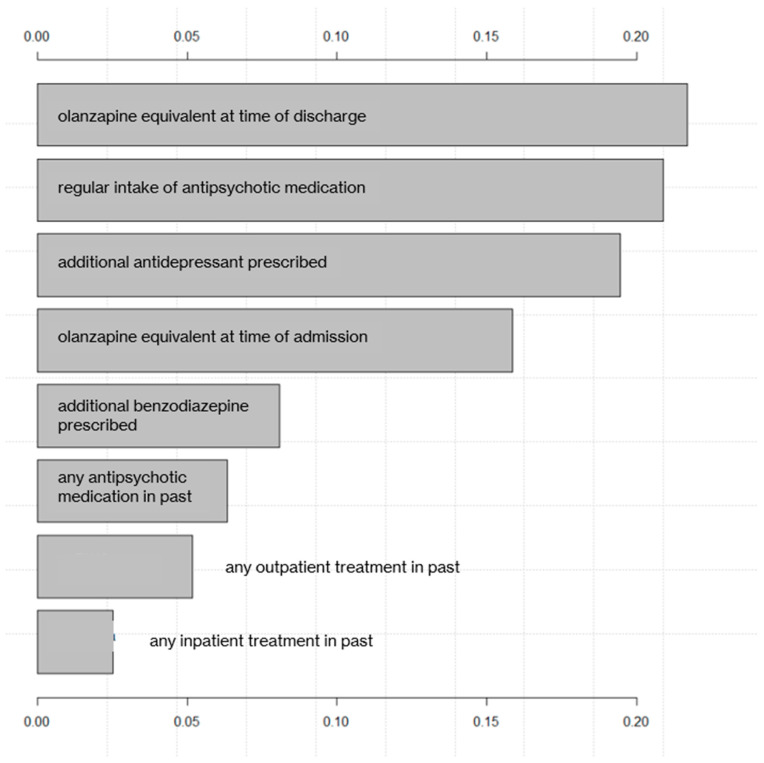
Ranking of predictor variables according to their relative influence.

**Table 1 biomedicines-10-03243-t001:** Absolute and relative distributions of predictor variables and levels of significance in NHST.

Variable Description	Offender Patients *n*/*N* (%)	Non-Offender Patients *n*/*N* (%)	*p*-Value *
Sociodemographic Data			
Age at admission (mean)	34.8 (10.5)	35.4 (11.2)	0.702
Gender: male	161/176 (91.5)	187/206 (90.8)	0.703
Country of birth, Switzerland	98/206 (47.6)	108/177 (61)	0.017 *
No school graduation (at admission)	128/189 (67.7)	75/161 (46.6)	0.000 *
Psychiatric Data			
Any outpatient psychiatric treatment in the past	115/196 (58.7)	141/159 (88.7)	0.000 *
Any inpatient psychiatric treatment in the past	152/200 (76)	174/177 (98.3)	0.000 *
Any antipsychotic medication in the past	134/206 (65)	165/173 (95.4)	0.000 *
Regular intake of antipsychotic medication	14/122 (11.5)	79/148 (54.4)	0.000 *
Comorbid alcohol use disorder	125/192 (65.1)	76/168 (45.2)	0.000 *
Comorbid substance use disorder	156/206 (75.7)	107/172 (62.2)	0.009 *
Comorbid personality disorder	31/206 (15)	14/150 (9.3)	0.183
Any compulsory measure in the past	100/184 (54.3)	62/142 (43.7)	0.089
Any compulsory measure currently	81/204 (39.7)	26/176 (14.8)	0.000 *
Length of stay (in weeks)	134.4 (124.7)	8.8 (7)	0.000 *
PANSS at admission	24.7 (12.8)	22.1 (10)	0.114
PANSS at discharge	12.2 (9.9)	12.9 (10.7)	0.834
Data on current medication			
Olanzapine equivalent at admission (mg)	21.4 (14.3)	14.6 (12.1)	0.000 *
Olanzapine equivalent at discharge (mg)	22.1 (12.3)	19.3 (14.2)	0.008 *
Polypharmacy ^1^ at admission	34/145 (23.4)	54/146 (37)	0.024 *
Polypharmacy ^1^ at discharge	72/206 (35)	72/178 (40.4)	0.333
Typical antipsychotic prescribed	37/204 (18.1)	24/177 (13.6)	0.188
Clozapine prescribed	77/204 (37.7)	54/177 (30.5)	0.192
Additional benzodiazepine prescribed	37/204 (18.1)	68/178 (38.2)	0.000 *
Additional antidepressant prescribed	18/204 (8.8)	63/178 (35.4)	0.000 *

* Statistical significance *p* < 0.05). ^1^ Polypharmacy was defined as a prescription of two or more antipsychotic substances.

**Table 2 biomedicines-10-03243-t002:** Machine learning models and their performance in cross-validation (nested resampling).

Scheme	Balanced Accuracy (%)	AUC	Sensitivity (%)	Specificity (%)	PPV (%)	NPV (%)
Logistic Regression	74.9	0.84	68.90	81.10	76.30	75
Tree	73.9	0.79	70.90	77.10	72.60	75.5
Random Forest	75.3	0.84	70.7	79.9	75.3	76.5
GradientBoosting	76.2	0.85	69.8	82.6	78.2	76
KNN	74.6	0.82	70.9	78.2	74.1	75.7
**SVM**	**77.8**	**0.87**	**77.8**	**77.9**	**74.2**	**79.3**
Naïve Bayes	77	0.85	76.5	77.6	74.5	79.5

AUC = area under the curve (level of discrimination); PPV = positive predictive value; NPV = negative predictive value; KNN = k-nearest neighbor; SVM = support vector machine; bold = indicator of best predictive model.

**Table 3 biomedicines-10-03243-t003:** Final SVM model performance measures.

Performance Measures	% (95% CI)
Balanced Accuracy	73.7 (65.6–81.1)
AUC	0.83 (0.76–0.90)
Sensitivity	66.7 (52.4–78.5)
Specificity	82.3 (70.1–90.4)
PPV	76.6 (61.6–87.2)
NPV	73.9 (61.7–83.4)

AUC = area under the curve (level of discrimination); PPV = positive predictive value; NPV = negative predictive value; SVM = support vector machine.

## Data Availability

The dataset generated and analyzed during the current study are available from the corresponding author upon reasonable request. A detailed list of all our variables (including definitions and references) is available under the following link: https://www.researchgate.net/publication/363044110_Coding_protocol_Pathways_into_delinquency_in_offenders_suffering_from_schizophrenia_spectrum_disorders, uploaded August 2022, last accessed on 6 December 2022.

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
