# Peer review of "High Risk, High Dose?—Pharmacotherapeutic Prescription Patterns of Offender and Non-Offender Patients with Schizophrenia Spectrum Disorder"

_biomedicines, 2022, doi:10.3390/biomedicines10123243_

Round 1

Reviewer 1 Report

I read this study with considerable interest and it seems to me that is more than worthy of the rapid presentation.

However, it seems to me that the group of the “offender” should be better defined, providing the possible criminal background. Possibly, they might be the subgroups that would contribute to better support of the suggested findings.  

Author Response

“However, it seems to me that the group of the “offender” should be better defined, providing the possible criminal background. Possibly, they might be the subgroups that would contribute to better support of the suggested findings.”

Thank you for your valuable contribution, which we have gladly implemented in our manuscript. We have introduced a paragraph to the methods section explaining the offences leading to the referenced forensic psychiatric hospitalization (see ll 66-69):

“Offences leading to the referenced forensic psychiatric hospitalization included both violent crime – (attempted) homicide, assault, violent offences against the sexual integrity, robbery, arson – and non-violent crime (threat and coercion, property crime without violence, criminal damage, traffic offences, drug offences and illegal gun possessions).”

As well as the following paragraph, which is supposed to describe the offender population selected for this specific study (see ll 87-91):

“Our final sample then consisted of 205 offender patients and 178 non-offender patients. Amongst the latter, 40.4% had committed a violent offence, including violent crimes against the sexual integrity, which accounted for 2.4% of all violent offences. The mean age at the first entry in the Swiss criminal registry was 24.7 years.”

Reviewer 2 Report

This is a well-structured research study aimed at testing whether offender patients receive higher doses of antipsychotic drugs, are more often subjected to antipsychotic polypharmacy, or receive benzodiazepines for sedation more frequently. To this end they studied we chose 370 male and female offender patients with a diagnosis of F2x according to ICD‐10, who had had all been in court mandated treatment at the Centre for Inpatient Forensic Therapies of the University Hospital of Psychiatry Zurich, Switzerland. The authors employed several machine learning methods to predict 

I have several concerns:

-       The sample size is large but probably not sufficiently powered for ML analysis. How did the authors account for overfitting? To what extent the nested resampling model is able to address this? Please provide the performance parameters.

-       The presence of diagnostic heterogeneity is not adequately addressed. F2x includes several diagnoses…

-       The variables that are identified as important predictors should be labeled clearly not with codes

-       The importance of the predictors is not adequately discussed

-       The Discussion should be focused first on the main results and then streamlined to compare the results with the literature available

Author Response

Thank you for your diligent and helpful revisions, which we highly appreciate. We have tried to implement your feedback in our revised manuscript, hopefully to your satisfaction. Please find your comments and our corresponding replies (in italics) below.

- "The sample size is large but probably not sufficiently powered for ML analysis. How did the authors account for overfitting? To what extent the nested resampling model is able to address this? Please provide the performance parameters."

Yes, the sample size is indeed rather small for ML purposes, and we want to be clear about this limitation in our discussion. We have added a short paragraph elaborating on the problem of overfitting and the nested resampling approach in the methodology section (see line 174ff). The performance of each model in the nested resampling validation is shown in table 2. Please let us know if you feel there are any other parameters that would be beneficial to the reader.

“As our sample size was relatively small, we were careful about avoiding overfitting, a common obstacle in ML occurring when e. g. outliers are incorporated in the model.  For this purpose, it is advisable to apply imputation, variable reduction and model building in a cross-validation process, and to keep it separate from the testing of the model. In our study, a nested resampling approach was employed, using a nested resampling model with the inner loop performing imputation, variable filtration, and model building within fivefold cross-validation, and the outer loop for performance evaluation also embedded in fivefold cross-validation.”

Furthermore, we have mentioned the issue of overfitting in the limitations paragraph of the discussion (see ll. 449-454).

-  "The presence of diagnostic heterogeneity is not adequately addressed. F2x includes several diagnoses…"

We have added a description of the diagnostic composition in both groups, which both predominantly were diagnosed with paranoid schizophrenia (F20.0 acc. to ICD.10), and we have included this aspect in the paragraph on limitations (see l 91 ff & 455 ff).

- "The variables that are identified as important predictors should be labeled clearly not with codes."

Thank you for bringing this to our attention, we absolutely agree. We have now adapted the figure with the predictors written out instead of the codes to improve readability and clarity (see Figure 2).

- "The importance of the predictors is not adequately discussed."

We have added another paragraph on possible explanations for the weight of each predictor variable in our discussion (see ll 411 ff)

“When ranking the identified predictors in accordance with their contributed weight to the ML model, the olanzapine equivalent at the time of discharge from the referenced hospitalization, as measure of the cumulative antipsychotic dose, emerged as most relevant in distinguishing offender and non-offender patients. The following five most powerful predictors also referred to psychopharmacologic treatment (regular intake of antipsychotic medication, additional prescription of antidepressants and benzodiazepines, a history of antipsychotic pharmacotherapy). While the two groups also significantly differed regarding their prevalence of psychiatric comorbidities, neither of these variables emerged as nearly as powerful regarding their predictive value in the model as pharmacological items – which is underlined by the fact that the AUC of the model did not improve by more than 5% through adding another item. The observation that prescription rather than clinical features dominated the model by far could be an expression of the different treatment settings, approaches, and goals between forensic and general psychiatric institutions.

-  "The Discussion should be focused first on the main results and then streamlined to compare the results with the literature available.”

We have changed the structure of the discussion, starting with a short summary of both objectives (a. testing the three hypotheses, b. ML model building) and then a presentation of the main results regarding the first objective. We have then focused on the findings regarding the first objective. However, we felt it difficult to further streamline the discussion as the two objectives were very intertwined. If you do feel that further restructuring and modifications to the discussion is necessary, we will of course gladly apply further changes as best as possible.

Round 2

Reviewer 2 Report

No further comments